# Clinical Evaluation of AMNIODERM+^®^ Wound Dressing Containing Non-Viable Human Amniotic Membrane: Retrospective-Perspective Clinical Trial

**DOI:** 10.3390/biotech13030036

**Published:** 2024-09-19

**Authors:** Iveta Schmiedova, Petr Slama, Alena Dembickaja, Beata Kozova, Vendula Hyneckova, Sona Gogolkova, Elen Stastna, Michal Zahradnicek, Stefan Savic, Arash Davani, Edward Hulo, Emil Martinka

**Affiliations:** 1BioHealing s.r.o., Dr. Slabihoudka 6232/11, 708 00 Ostrava, Czech Republic; 2Laboratory of Animal Immunology and Biotechnology, Department of Animal Morphology, Physiology and Genetics, Faculty of AgriSciences, Mendel University in Brno, Zemedelska 1, 613 00 Brno, Czech Republic; 3National Institute of Endocrinology and Diabetology, Kollárová 282/3, 034 91 Ľubochňa, Slovakia

**Keywords:** medical device, Vaselinum album, tulle dressing, grassy tulle, amniotic membrane, chronic wounds, non-healing wounds

## Abstract

Chronic wounds result from the body’s inability to heal, causing pain, pathogen entry, limited treatment options, and societal burden. Diabetic foot ulcers are particularly challenging, often leading to severe complications like leg amputation. A clinical study tested AMNIODERM+^®^, a new device with a lyophilized human amniotic membrane (HAM), on chronic diabetic foot ulcers. Participants had diabetic neuropathic or neuroischemic leg wounds (2–16 cm^2^) unhealed by 20% after six weeks of standard care. This study showed significant wound healing improvements with AMNIODERM+^®^. The median wound size reduction after 12 weeks was 95.5%, far exceeding the null hypothesis of 20% change. Additionally, 65% of patients achieved complete ulceration healing, surpassing the 50% efficacy requirement. The median time to full closure was 11.4 weeks, with the proportion of completely healed patients rising progressively, reaching 55% by week 11. These findings, from the clinical trial “Freeze-dried amniotic membrane in the treatment of nonhealing wounds”, suggest AMNIODERM+^®^ as a promising future treatment for chronic diabetic foot ulcers. The published results were obtained as part of a clinical trial entitled “Freeze-dried amniotic membrane in the treatment of nonhealing wounds: a single-arm, retrospectively-perspective clinical trial”, EUDAMED Nr. CIV-SK-22-10-041146.

## 1. Introduction

Diabetes mellitus encompasses a cluster of metabolic conditions marked by elevated blood glucose levels. People with diabetes face a higher likelihood of encountering severe, potentially life-threatening health complications, leading to escalated healthcare expenses, diminished quality of life, and heightened mortality rates [1]. Chronically elevated blood glucose levels lead to widespread vascular damage that impacts the heart, eyes, kidneys, and nerves, giving rise to a range of complications [2]. Prolonged hyperglycemia can lead to a range of complications, with diabetic foot ulcers (DFUs) being the primary contributor to amputation and disability [3]. As per the International Diabetes Federation’s data, the global diabetic population aged 18 years and older amounted to 451 million individuals in 2017 [4]. DFUs are a major challenge for clinical practice because they are not always easy to treat with standard methods and can cause serious complications, including leg amputation [5]. Calluses, blisters, cuts, burns, and ingrown toenails have the potential to progress into diabetic foot ulcers. Peripheral neuropathy can result in patients not being able to detect these minor injuries promptly, allowing ulcers to form and grow unnoticed [6]. Currently, the conventional approaches to manage DFUs involve maintaining tight control over blood sugar levels, meticulous debridement, employing moisture-retaining dressings, relieving pressure on weight-bearing ulcers, assessing leg circulation, and administering antibiotics in case of infection [7]. However, despite these proactive measures, numerous DFUs continue to pose significant challenges in terms of achieving complete healing [8]. While there are various methods of treating DFU, advanced methods such as negative pressure wound therapy, hyperbaric oxygen therapy, and biological dressings are becoming increasingly popular. These biological dressings include amniotic membranes, which have positive effects on the healing process and can improve the prognosis of patients with chronic wounds [9]. In an earlier study, we evaluated an amniotic membrane-based product that is classified as a tissue graft under the commercial brand Amnioderm^®^. This study demonstrates the positive effect of human amniotic membrane on the healing of chronic wounds [10].

AMNIODERM+^®^ is a new medical device that uses lyophilized human amniotic membrane and is designed for the treatment of chronic DFUs unresponsive to standard treatment. This clinical trial investigated the efficacy and safety of AMNIODERM+^®^ in patients with chronic DFUs who did not respond to standard treatment. The aim of the study was to assess whether AMNIODERM+^®^ can be an effective and safe treatment option for these patients.

Chronic wounds are associated with the impairment of the following processes: growth factor production, angiogenic response, macrophage differentiation, collagen production, epidermal barrier function, granulation tissue formation, and keratinocyte and fibroblast migration/proliferation. Decreased cell migration results in deficient re-epithelialization of the chronic wound [11,12].

Re-epithelialization involves multiple processes, including the formation of a provisional wound bed matrix, the migration of epidermal keratinocytes from the cut edge of the wound, the proliferation of keratinocytes that feed the advancing and migrating epithelial tongue, the stratification and differentiation of the new epithelium, and the reformation of the basement membrane zone. The elevation of metalloproteinases excessively degrades the local extracellular matrix and thus impairs cell migration. Competition between inflammatory and anti-inflammatory signals lead to a misbalanced environment for proper wound healing to occur. The presence or absence of certain extracellular matrix molecules such as collagen, laminin, and fibronectin within any basement membrane have a huge influence on the adhesion and growth of the overlying stem cells. As well as allowing the cells to attach and migrate, the extracellular matrix molecules also serve as adhesion ligands, which transmit signals via their interactions with cell surface receptors [13,14,15].

Diabetes mellitus affects wound healing through altered protein and lipid metabolism, and thereby affects abnormal granulation tissue formation (Figure 1). Increased glucose levels in the body lead to the uncontrolled covalent bonding of aldose sugars to a protein or lipid without any normal glycosylation enzymes. Re-epithelialization is delayed in diabetic wounds due to impaired keratinocyte and fibroblast functions brought on by hyperglycemia and the accumulation of advanced glycation end-products [16].

Ideal wound dressing is supposed to help faster re-epithelialization and collagen synthesis, promoting angiogenesis by creating hypoxia to the wound bed and decreasing wound bed pH, which leads to a decrease in the wound infection.

The AMNIODERM+^®^ (BioHealing s.r.o., Ostrava, Czech Republic) wound dressing is an advanced wound care medical device that acts as a mechanical barrier to protect the wound. It acts as a physical barrier against bacterial contamination and also creates the moist environment required for healing. It supports the creation of migration tongue, maintains homeostasis, creates a barrier for the wound, and retains moister balance. This wound dressing is designed to maintain contact with irregular shapes and surfaces, providing a flexible and gently adherent covering for the surface of various wounds.

To test the efficacy of this medical device, a retrospective-prospective clinical study was conducted. This type of study involves looking back at previous medical records and analyzing the outcome of patients who had used various SoC treatment for their non-healing wounds, and prospectively evaluating these wounds after the use of the dressing. The rationale behind this study was to gather data on the effectiveness of this AMNIODERM+^®^, including its safety, tolerability, and efficacy.

This was a retrospective-prospective clinical study. The retrospective data included information on the subject’s medical history and standard of care (SoC) wound treatment. The prospective data contained information on wound treatment with AMNIODERM+^®^.

The clinical trial was planned without a control or comparison group. A control group was not considered necessary because the intent of the clinical trial was to investigate the percentage reduction in wound area at 12 weeks in patients with a diabetic neuropathic or neuroischemic wound anywhere on the leg who have been treated with a long-term SoC resulting in a healing effect of less than 20% of the wound area. Data from a 6-week period of SoC treatment were collected retrospectively.

## 2. Materials and Methods

### 2.1. New Medical Device AMNIODERM+^®^

The wound dressing is a single-use, multilayer, sterile medical device. It is lyophilized, solid, and semi-transparent, and can be stored at temperatures between 15 and 25 °C for up to five years.

The dressing is classified as medical device class III, as it contains human tissue and this increases its risk class.

This dressing is designed for single application and is intended for hard-to-heal wounds. Application is performed by medical staff (physicians and nurses in outpatient and hospital care) or by adult patients themselves, or adults in their immediate circle. Each package contains one sterile piece, intended for immediate use after opening. Multiple dressings can be applied to a single wound. The dressing may be left in place for up to 7 days, depending on the wound and surrounding skin condition.

The dressing provides physical conditions to support the natural healing process in various hard-to-heal wounds, such as partial and full-thickness wounds, pressure ulcers, venous ulcers, diabetic ulcers, chronic vascular ulcers, surgical wounds, and trauma wounds. It acts as a physical barrier to protect the wound and help prevent bacterial contamination, maintaining a moist environment critical for effective healing. The gentle inner layer prevents damage to newly formed cells during repositioning or removal. The contact layer is made of Vaselinum album. The dressing has a unique multi-layer design, as shown in Figure 2:

### 2.2. Mechanism of Action

AMNIODERM+^®^ is intended to be used for providing mechanical barrier in hard-to-heal wounds of various etiologies such as partial and full-thickness wounds, pressure ulcers, venous ulcers, diabetic ulcers, chronic vascular ulcers, surgical wounds, and trauma wounds. The wound dressing acts as follows:Protection from external environment.Prevention of liquid evaporation.Prevention of microorganisms from entering the wound.

All of these features work together in synergy to create an optimal homeostatic environment in the wound, which results in faster healing of hard-to-heal wounds. The gentle inner side prevents damage to the newly formed cells during repositioning or device removal.

This innovative wound dressing incorporates three key components: polyester fibers, petroleum jelly/Vaselinum album, and a non-viable amniotic membrane (Table 1), each contributing to distinct aspects of wound management. The wound dressing aims at facilitating effective wound management through purely mechanical means.

### 2.3. Method of Conducting the Study and Its Evaluation

The clinical trial methodology is summarised in the Table 2 below:

## 3. Results

### 3.1. Primary Endpoint

Median change in wound size after 12 weeks was 95.5% (minimum to maximum: 47.7–99.6%). This was significantly more compared to the null hypothesis of 20% wound change (*p* < 0.001; one sample Wilcoxon signed rank test).

### 3.2. Secondary Endpoints

Wound closure after 12 weeks was observed in 13 patients, which equals 65.0% (95% CI: 40.8–84.6%). Compared to the null hypothesis that wound closure would be observed in 50% of patients, there was no significant difference (*p* = 263; one-sample exact binomial test; Table 3).Time to complete wound closure was 11.4 weeks (95% CI: 8.3 months—upper confidence limit not reached).The proportion of patients with complete wound closure each week is described in Table 3.Subject satisfaction and user assessment at the final visit is described in Table 4.

Treatment with AMNIODERM+*^®^* was effective and efficient. Both IIT (intention-to-treat) and PP (per-protocol) results analyses demonstrated high treatment efficacy. In the primary endpoint (% reduction in ulceration area after 12 weeks of treatment), there was a significant reduction in ulceration area with a median reduction of up to 95.5% (*p* < 0.001), which was significantly more than the ≤20% threshold considered to be ineffective treatment. The secondary endpoint (proportion of patients with complete ulceration healing) was achieved in 65% (95% CI: 40.8–84.6%) of patients, which was when the 50% requirement set by the protocol (based on international definitions of treatment efficacy) was met. Among other secondary endpoints, the median time to complete ulceration closure was 11.4 weeks. The proportion of patients with complete closure of ulceration each week increased progressively until week 10 from 10% at week 2 to 25% at week 10, increasing significantly to 55% at week 11 ([Fig biotech-13-00036-ch001]).

Figure 3 and Figure 4 below show examples of complete wound closure in two patients.

The investigator asked the subject on adverse events, serious adverse events, and incidents at each visit, and also examined the subject to confirm/exclude possible AEs. All safety information was recorded by an investigator in the eCRF (adverse event form, pregnancy form, and device deficiencies form). No safety information was observed during the investigation. No adverse events or device deficiency were observed during the study.

## 4. Discussion

The human amniotic membrane (HAM) consists of three primary components: active cells, collagen fibers, extracellular matrix, and regenerative molecules. Researchers have extensively examined the amniotic membrane to understand its impact on the wound healing process [18]. In a study by DiDomenico et al. in 2016, it was revealed that the average and middle time required for wound healing was 12 weeks when utilizing the dehydrated human amnion and chorion allograft. This timeframe proved to be swifter compared to the majority of other cellular or tissue-based products documented in various randomized clinical trials [19]. In the past, the amniotic membrane has been noted to possess numerous soluble growth factors and cytokines, many of which are linked to diverse stem cell functions. Despite research showing that the surface structure of the amniotic membrane can affect the behavior of stem cells, the primary factors responsible for the clinical variability have not been pinpointed [20]. From a mechanical perspective, this tissue exhibits remarkable flexibility and resilience, enduring the gradual expansion during embryonic growth and shielding it against external injuries [21]. Grémare et al. in 2019 demonstrate that the human amniotic membrane can indeed serve as a preferred raw material, particularly in the advancement of tissue-based products where mechanical characteristics play a key role [22].

In scientific papers, the usual onset of treatment with products containing amniotic membranes is between the 6th and 8th week of treatment. A similar outcome with the use of HAM for the treatment of DFU was also observed in a clinical study conducted by the MiMedx Group (Marietta, GA, USA). In this study, a 40% wound reduction was observed after 4 weeks of treatment. Another study showed 92% wound closure after 6 weeks of treatment in wounds 4–51 weeks old. A study from Osiris Therapeutics demonstrated complete wound closure by 12 weeks in 62% of treated patients [23]. In studies, healing of the studied wounds can be observed within 6–12 weeks, demonstrating the effectiveness of amniotic membrane treatment compared to standard care.

Our clinical trial with a medical device containing a non-viable amniotic membrane had a success rate of 65%. Here, there was a delayed onset of effects between weeks 10 and 11 of treatment.

This delayed onset of treatment with the medical device containing the HAM component compared to the stand-alone HAM is likely due to the fact that this medical device does not release cytokines and growth factors into the wound as it does with the stand-alone HAM. By not having this supportive function, the HAM has a covering function and a function where it creates a natural homeostatic environment in the wound and the patient’s body, and then closes the wound naturally without supportive agents. This method of treatment has a slower onset of action, but a longer-lasting effect.

The undeniable advantage of treating non-healing wounds with AMNIODERM+^®^ is the availability and high efficacy of this medical device. These two aspects combine into a low-cost but affordable SoC and a highly effective but expensive advanced therapy available only in specialized centers.

AMNIODERM+^®^ provides a solution for wound epithelialization. It is at this stage that wounds very often stagnate for several years [23,24]. There are no commonly available products for this phase of wound healing.

In this study, the potential of the mechanical properties of the amniotic membrane to support the wound dressing function of a medical device was tested. The primary objective of this study was to assess the percentage reduction in wound area after a 12-week period of treatment with AMNIODERM+^®^ wound dressing. The results revealed a significant reduction in wound size during the treatment duration. The median percent change in wound size after 12 weeks was 95.5%, ranging from 47.7% to 99.6%. This reduction was notably higher than the null hypothesis of a 20% wound change (*p* < 0.001; assessed using the one sample Wilcoxon signed rank test, as shown in Table 2).

The secondary endpoint of this study focused on the proportion of subjects achieving complete wound closure after 12 weeks of treatment with AMNIODERM+^®^. The findings indicate that 65% of patients achieved complete wound closure, with a 95% confidence interval ranging from 40.8% to 84.6% (as presented in Table 3). This result did not exhibit a statistically significant difference from the expected 50% closure rate. To establish statistical significance for closure in more than 50% of patients after 12 weeks of AMNIODERM+^®^ treatment, a larger patient sample size would be necessary.

The median time required for complete wound closure was 11.4 weeks with a 95% confidence interval ranging from 8.3 weeks to the upper limit, which was not reached due to the proximity of the median time to the end of the observation period, as displayed in Table 4. The proportion of patients with closure steadily increased up to week 10 (from 10% at week 2 to 25% at week 10), and then notably rose to 55% at week 11. The percentage reduction in the wound area was evaluated weekly. The wound size demonstrated a significant decrease after the first week of applying AMNIODERM+^®^, with a median percentage change of 45% (as indicated in Table 2). Over the treatment period, the wound size gradually diminished, with a median reduction of 95% at both weeks 11 and 12.

The Numeric Rating Scale (NRS-11) was employed to gauge pain intensity over the 12-week observation period. The NRS-11 scores remained relatively stable, with a minor change from a mean value of 1.3 at baseline to 1.0 at week 12 and a consistent median value of 1 throughout the study (as shown in Table 4). The Foot Health Status Questionnaire (FHSQ) consisted of nine domains assessing various aspects of foot health. While the values of all domains did not exhibit significant changes during the observation period, they remained stable (Table 4).

User satisfaction and assessment were evaluated at the final visit. A survey revealed that 70% of patients completely agreed and 30% agreed that the application process of AMNIODERM+^®^ was manageable and did not cause significant discomfort. User assessment of AMNIODERM+^®^ encompassed three questions. About 85% of users expressed great satisfaction with the management of AMNIODERM+^®^, while 15% were satisfied. Additionally, 70% of users reported an excellent experience with AMNIODERM+^®^ treatment and 100% perceived an excellent improvement in wound condition after using the product. These results underscore the positive reception and satisfaction among patients to whom a AMNIODERM+^®^ has been applied.

## 5. Conclusions

In conclusion, this study demonstrates that AMNIODERM+^®^ is a safe and highly effective treatment for chronic diabetic neuropathic or neuroischemic ulcerations (DFUs) that are unresponsive to standard care (SoC). The primary endpoint, a significant reduction in ulceration area after 12 weeks, was achieved with a median reduction of 95.5% (*p* < 0.001), well above the ineffective treatment threshold of ≤20%. Additionally, 65% of patients experienced complete ulcer healing, surpassing the required 50%. The treatment showed a gradual and consistent healing trend, with a median time to complete closure of 11.4 weeks. Importantly, no adverse events were recorded, and patients found the treatment to be comfortable. Unlike conventional amniotic membranes, AMNIODERM+^®^ works through a mechanical mechanism, creating a stable environment for natural wound healing, which, while slower to take effect, provides a longer-lasting result. These findings highlight the potential of AMNIODERM+^®^ as a valuable treatment option in routine clinical practice for managing chronic DFUs. It was shown that treatment of non-healing wounds with AMNIODERM+^®^ is significantly more effective than treatment with SoC. At the same time, the new medical device was found to have higher efficacy and effectiveness compared to the conventional amniotic membrane when compared to previously conducted studies by the sponsor. The onset of the treatment effect with the conventional amniotic membrane is faster, and the likely mechanism of action is the release of different types of bioactive molecules into the wound [25].

## Data Availability

The published results were obtained as part of a clinical trial entitled: “Freeze-dried amniotic membrane in the treatment of nonhealing wounds: a single-arm, retrospectively-perspectiveclinical trial”, EUDAMED Nr. CIV-SK-22-10-041146. DOPLNIT ODKAZ NA STUDII.

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
