# Peer review of "Clinical Evaluation of AMNIODERM+® Wound Dressing Containing Non-Viable Human Amniotic Membrane: Retrospective-Perspective Clinical Trial"

_biotech, 2024, doi:10.3390/biotech13030036_

Round 1

Reviewer 1 Report

Comments and Suggestions for Authors

1. The Conclusion section needs to be clear and concise. The significant results and primary conclusions of the paper should be emphasized and expressed more precisely.

2. The article contains several formatting errors. For instance, ensure the spelling of references aligns with the journal's style (e.g., Reference 22). Carefully review and correctly use abbreviations.

3. Enhance the background information on wound healing by citing 10.1002/adma.202306632 and 10.1016/j.mtbio.2022.100264. Also, outline the advantages of the current research compared to previously published articles in related areas.

4. Avoid using abbreviations in the Abstract unless they are frequently used (such as HAM). The Abstract should clearly and simply convey the article's topic to the reader.

Author Response

COMMENTS 1: The Conclusion section needs to be clear and concise. The significant results and primary conclusions of the paper should be emphasized and expressed more precisely.
RESPONSE 1: Edited in article

COMMENTS 2: The article contains several formatting errors. For instance, ensure the spelling of references aligns with the journal's style (e.g., Reference 22). Carefully review and correctly use abbreviations.
RESPONSE 2: Edited in article

COMMENTS 3: Enhance the background information on wound healing by citing 10.1002/adma.202306632 and 10.1016/j.mtbio.2022.100264. Also, outline the advantages of the current research compared to previously published articles in related areas.
RESPONSE 3: The authors of the article decided not to use the above mentioned publications in the preparation of the article, because they are rather marginally related to the described topic.

COMMENTS 4: Avoid using abbreviations in the Abstract unless they are frequently used (such as HAM). The Abstract should clearly and simply convey the article's topic to the reader.
RESPONSE 4: Edited in article

Reviewer 2 Report

Comments and Suggestions for Authors

A clinical study tested AMNIODERM+®, a new device with lyophilized human amniotic membrane (HAM), on chronic DFUs. Participants had diabetic neuropathic or neuroischemic leg wounds (2-16 cm²) unhealed by 20% after six weeks of standard care. The study showed significant wound healing improvements with AMNIODERM+®.

1. In the wound structure, is the vaselinum album active ingredient for promoting the wound healing? IF yes, how many the thickness of the vaselinum album? how to control it? Additionally, which materials is the non-viable amniotic membrane?

2. Page 10, it is Table 4, not Table 1.

3. Conclusions should be more concise.

Author Response

COMMENTS 1: In the wound structure, is the vaselinum album active ingredient for promoting the wound healing? IF yes, how many the thickness of the vaselinum album? how to control it? Additionally, which materials is the non-viable amniotic membrane? – RESPONSE 1: Vaselinum album is not an active layer, its function is structural and therefore provides the described properties. The Vaselinum album layer thickness is BioHealing's know-how and cannot be disclosed due to the planned commercial use of the described product.

COMMENTS 2: Page 10, it is Table 4, not Table 1.

RESPONSE 2: Edited in article

COMMENTS 3: Conclusions should be more concise.

RESPONSE 3: Edited in article

Reviewer 3 Report

Comments and Suggestions for Authors

In this manuscript, the authors present their clinical evaluations of a commercial rmedical device containing human amniotic tissue intended for treatment of chronic wounds, particulary in diabetic foot ulcers. The new medical device -AMNIODERM- was described, as well its mechanism of action. In fact, the article presents the results of a retrospectively-prospective clinical trial concerning AMNIODERM (retrospective data - with information about the subject's history and wound treatment (standard care); prospective data - with information about wound treatment with the new device.

The study presents in detail the clinical trial - as it was proposed and approved (as non-randomized, monocentric, single-arm study). The trial involved 20 patients who have followed the treatment for 12 weeks (without a control or comparison group). The findings indicated the efficaccy of the new wound dressing - 65% success rate (complete wound closure!) and good compliance (85% of users expressed great satisfaction), which argues the recommendation for its use.

In the context, the work is interesting (as topic and results) and the paper could be considered for publication in the journal.

 Comments and suggestion:

1. I did not understand why the authors did not indicate as reference another study published by the same team, on the same topic:  Schmiedova I, Ozanova Z, Stastna E, Kiselakova L, Lipovy B, Forostyak S. Case Report: Freeze-Dried Human Amniotic Membrane Allograft for the Treatment of Chronic Wounds: Results of a Multicentre Observational Study. Front Bioeng Biotechnol. 2021;9:649446. doi: 10.3389/fbioe.2021.649446.

I considered that it is important to clarify this aspect. The authors should cite the previous publication regarding the same medical device, and clearly present their novelty and difference compared to this one.

2.   At  2.Materials:  The authors should specify and argue the inclusion of AMNIODERM in the medical device risk class (class III) 

 Minor remark:

- Line 157 ???

- The reference format:  please check carefully.

Author Response

COMMENTS 1: I did not understand why the authors did not indicate as reference another study published by the same team, on the same topic:  Schmiedova I, Ozanova Z, Stastna E, Kiselakova L, Lipovy B, Forostyak S. Case Report: Freeze-Dried Human Amniotic Membrane Allograft for the Treatment of Chronic Wounds: Results of a Multicentre Observational Study. Front Bioeng Biotechnol. 2021;9:649446. doi: 10.3389/fbioe.2021.649446.

I considered that it is important to clarify this aspect. The authors should cite the previous publication regarding the same medical device, and clearly present their novelty and difference compared to this one.

RESPONSE 1: Edited in article

COMMENTS 2: At  2.Materials:  The authors should specify and argue the inclusion of AMNIODERM in the medical device risk class (class III)

RESPONSE 2: Edited in article

Minor remark:

- Line 157 ??? – We don't understand this comment

- The reference format:  please check carefully. Edited in article